# Effects of Genotype and Growing Year on the Nutritional Composition and Pasting Properties of Glabrous Canary Seed (*Phalaris canariensis* L.) Groat Flours

**DOI:** 10.3390/foods13111786

**Published:** 2024-06-06

**Authors:** Lovemore Nkhata Malunga, Sijo Joseph Thandapilly, Pierre J. Hucl, Nancy Ames

**Affiliations:** 1Agriculture and Agri-Food Canada, 196 Innovation Drive, Winnipeg, MB R3T 6C5, Canada; sijo.joseph@agr.gc.ca; 2Richardson Centre for Food Technology and Research, University of Manitoba, 196 Innovation Drive, Winnipeg, MB R3T 6C5, Canada; 3Department of Food and Human Nutritional Sciences, University of Manitoba, Winnipeg, MB R3T 6C5, Canada; 4Crop Development Centre, University of Saskatchewan, Saskatoon, AB S7N 5A2, Canada; pierre.hucl@usask.ca

**Keywords:** Glabrous Canary seed, hairless Canary seeds, nutritional properties, nutritional components, pasting properties, genotype by environment

## Abstract

Canary seed flour is a new food ingredient that the United States Food and Drug Administration (FDA) and Health Canada recently granted Generally Regarded as Safe (GRAS) status. Stability in nutritional composition and functional properties is an essential characteristic of food ingredients for consistency in nutritional quality and performance in processing. This work assessed the effect of genotypic and environmental variation on the nutritional (protein, starch, amylose, oil, dietary fiber, minerals and fat-soluble vitamins) and pasting (as measured in viscosity (peak, trough, breakdown, final, and setback), peak time, and pasting temperatures) properties of Canary seed. The samples included four Canary seed varieties grown in randomized complete block design experiments at one location for two growing seasons. In general, the nutritional composition of Canary seed flour was not affected by genotype, growing year, and their interaction except for starch content, which was significantly affected by the growing year (*p* < 0.0001), and iron content, which was affected by genotypic variation (*p* < 0.0001). The pasting properties of Canary seed flour were significantly (*p* < 0.001) affected by both genotypic and growing year variation but not their interaction. Our results suggest that the food industry should measure starch and iron content prior to processing to ensure consistency in nutritional labeling. Also, for those applications where starch pasting properties are essential, the manufacturer should consider measuring the RVA pasting viscosities for every batch of raw material. The results have provided the baseline knowledge of which nutritional or functional properties of Canary seed flour can be improved through breeding and agronomy programs to ensure the reliability of Canary seed as an ingredient.

## 1. Introduction

Glabrous Canary seed (*Phalaris canariensis* L.) is a novel cereal grain with Generally Regarded as Safe (GRAS) status [1,2] and provides a new plant food source for human consumption. A toxicology assessment demonstrated that it is safe for humans to consume glabrous Canary seed [3]. On the other hand, hairy (pubescent) Canary seed is unsafe for human consumption due to the potential health risk associated with its trichomes or spicules [4,5]. Thus, pubescent Canary seed grain is mostly grown for the caged bird feed market [6]. Glabrous Canary seed flour has potential for applications in baking, pasta production, bars, and beverages [6,7].

The nutritional composition of Canary seed flour has been reported previously [5,6,8]. Canary seed contains approximately 22% protein (which is higher than most cereal grains), 8% oil (similar to oats), 57% starch (similar to other cereal grains), and 6% total dietary fiber (lower than most cereal grains) [9]). Canary seed has a higher concentration of minerals than wheat and oats [8]. Starch pasting properties of Canary seed have been reported for both flour and isolated starch granules [10,11]. The nutritional or functional properties of cereal grains vary with genotype, environment, and their interaction [12,13,14]. However, only limited data are available on the effect of genotype and environment on the nutritional or functional properties of Canary seed.

Food manufacturers look for stability in nutritional composition and technological and functional properties when sourcing ingredients. The Codex Alimentarius Guidelines for Nutrition Labelling require the food industry to include “*energy value; and the amounts of protein, available carbohydrates (i.e., dietary carbohydrate excluding dietary fibre), fat, saturated fat, sodium, and total sugars; and the amount of any other nutrient for which a nutrition or health claim is made; the amount of any other nutrient considered to be relevant for maintaining a good nutritional status, as required by national legislation or national dietary guidelines*” [15]. Different government jurisdictions have adapted this Codex guideline in part or whole into their legislation. Thus, the food industry preprints labels for their food packaging with nutrition information. Therefore, food industries ensure that ingredients contain a desirable range of nutritional components in their quality control/management program. Also, the sensory quality of the final food product is dependent on the quality of the ingredients used. Thus, understanding factors affecting Canary seed flour’s nutritional and functional properties is critical for its successful utilization as a novel ingredient.

The main objective of this work was to assess the effect of genotypic and environmental variation on the nutritional and pasting properties of Canary seed. Thus, we studied the nutritional and pasting properties of four Canary seed varieties grown in randomized complete block design experiments at one location for two growing seasons.

## 2. Materials and Methods

### 2.1. Canary Seed Samples

Four hairless Canary seed varieties from the Crop Development Centre (CDC), University of Saskatchewan were used, which include two brown lines, CDC Calvi and CDC Lumio, and two yellow lines, CDC Cibo and C10045. These four Canary seed varieties were grown in three-replicate randomized complete block design (RCBD) experiments at one location in Saskatchewan (Kernen Crop Research Farm) over two years (2018 and 2019 growing seasons). The Canary seed samples were dehulled using a Codema Laboratory Huller LH 5095 system (Codema LLC, Minneapolis, MN, USA) [16]. The groats were cleaned and milled on a Retsch Ultra Centrifugal Mill ZM 200 (Retsch GmbH, Haan, Germany) using a 0.5 mm sieve. All salts, acids, and organic solvents used were bought from Fischer Scientific (Whitby, ON, Canada). All standards and enzymes used were bought from Millipore Sigma (Oakville, ON, Canada) unless stated otherwise. Megazyme assay kits were bought from Megazyme International Ireland (Bray, Wicklow, Ireland).

### 2.2. Macronutrient Composition Analysis

The American Association of Cereal Chemists (AACC) international standard methods were performed on cleaned groats ground into flour to determine moisture, protein, total starch, beta-glucan, and total dietary fiber contents [17,18,19,20]. Briefly, the percent of flour moisture was determined using oven drying for one hour at 130 °C [17]. Protein measurement followed a combustion method [18] using the Flash 2000 Elemental (N) Analyzer (ThermoFisher, Whitby, ON, Canada) where nitrogen was converted to protein by a factor of 6.25. Total starch and beta-glucan contents were measured by enzymatic/spectrophotometric assays [19,20] using the Megazyme kits K-AMYL and K-BGLU, respectively. The proportion of amylose in the starch was also determined following the Megazyme K-AMYL kit method. Crude fat was determined gravimetrically after acid hydrolysis according to the AOAC official method 922.06 [21].

### 2.3. Dietary Fiber Composition Analysis

Total (insoluble, soluble dietary, and low molecular weight) dietary fiber composition was analyzed according to the AACC method 32-50.01 [22] using the Ankom Fiber Analyzer (Ankom Technology, New York, NY, USA) and Knauer HPLC system with RI detector (K-2301) (Knauer, Berlin, Germany) equipped with a Waters Sugar-Pak^®^ (6.5 × 300 mm) (Waters Corporation, Milford, MA, USA) column for determining low molecular weight fiber.

Non-starch polysaccharides (NSP) were analyzed using the method of Englyst and Cummings [23] with some modifications [24]. Component sugars were analyzed by Gas Chromatography using a Flame Ionization Detector (GC-FID) (Agilent 6890 N gas chromatography system (GC) (Agilent Technologies, Santa Clara, CA, USA)), and uronic acids were analyzed by colorimetry.

### 2.4. Micronutrient Composition Analysis

Mineral contents were determined using the Inductively Coupled Plasma-Optical Emission Spectrometry (Agilent 7850 ICP-MS) method after microwave-assisted acidic digestion (CEM Mars 6 Microwave, Matthews, NC, Canada) as described by Malunga et al. [25]. Vitamins A (as retinol) and E (as alpha-tocopherol equivalent) were analyzed using the AFVAN-SLMF-0013 method at an accredited food testing laboratory (SGS Canada Inc., Burnaby, BC, Canada).

### 2.5. Soluble Mono-, Di-, and Oligo-Saccharide Composition Analysis

Soluble mono-, di-, and oligo-saccharide were extracted, concentrated, and derivatized using the Slominski et al. method [26]. Briefly, 200 mg of Canary seed flour was mixed with 6 mL aqueous ethanol (80%) containing 5 mg of internal standard (myo-inositol (Sigma, Burlington, MA, USA, I5125-500 G). The mixture was incubated for 18 h at room temperature under continuous mixing (magnetic stirring bar) and centrifuged (3000 rpm, 10 min at room temperature). The supernatant (1 mL) was transferred to a 2 mL microfuge tube and dried under vacuum (40 °C). Derivatization reagent (500 µL) was added and incubated at 40 °C for 1 h with continuous mixing (1000 rpm). Derivatization reagent constituted 2 acetone: 1 N,O-Bis(trimethylsilyl)acetamide, (Fisher, AC156461000): 0.1 Chlorotrimethylsilane, (Sigma, 386529-100 ML): 0.05 1-methylimidazole)**.** External standards including fructose (Sigma, F0127-500 G), glucose (Sigma, G8270-100 G), sucrose (Sigma, S7903-250 G), maltose (Sigma, 63418-25 G), raffinose (Sigma, R0250-25 G), and stachyose (Sigma, S4001-1 G) were also derivatized as described above to establish the response factor and elution time. The derivatized sample (1 μL) was injected into an Agilent 6890 N gas chromatography system (GC) (Agilent Technologies, Santa Clara, CA, USA) equipped with a flame ionization detector by using an auto-sampler (Agilent Technologies 7863 B series). The sample was separated on a Restek capillary column RXI-5 MS 30 m × 0.25 mm × 0.25 μm using helium gas (4 mL/min). The GC conditions were set as follows: injection (300 °C), column (initial 160 °C and ramp 10 °C per minute until 220 °C, hold 4 min, ramp 10 °C per minute until 340 °C, hold for 8 min), detector (300 °C, airflow 30 mL/min, H_2_ flow 35 mL/min and makeup flow of 30 mL/min) and a total run time of 30 min. Simple sugars were identified by using the retention times of their respective standards and a response factor of myo-inositol was used for quantification.

### 2.6. Pasting Properties of Canary Seed Flours

The pasting properties of Canary seed flour in water (25 g water + 3.5 g flour corrected to 14% moisture) were assessed using the Rapid Visco Analyzer (RVA 4500, Perten Industries, Stockholm, Sweden). The standard pasting 13 min profile (STD1) was used as per the AACC Approved Method 76-21.02 [27], which heats the slurry to 95 °C, holds for 2.5 min then cools back down to 50 °C with constant stirring at 160 rpm. Pasting parameters from the resulting viscosity (cP) over time curve were recorded, including peak (during 95 °C stage), trough, breakdown (peak minus trough), setback (final minus trough), and final viscosities.

### 2.7. Color Measurement of Canary Seed Flour and Groats

Groat and flour color (L*, a*, and b* values) were measured using a Minolta Chroma Meter CR-410 with the granular materials attachment CR-A50 (model CR-410, Minolta Co., Osaka, Japan)

### 2.8. Statistical Analysis

One-way analysis of variance (ANOVA) was performed using a generalized linear model procedure using SAS 9.4 software (SAS Institute Inc., Cary, NC, Canada) to study the effect of genotype and growing year on the nutritional and functional properties of Canary seed. Sample means were compared at *p* ≤ 0.05 by following the Tukey–Kramer honest significant difference method.

## 3. Results and Discussion

### 3.1. Macronutrient Composition

Table 1 presents the macronutrient composition of Canary seed flour. Protein content ranged from 22.65 to 23.9% db, which is higher than the protein content reported for other cereal grains (wheat (15–19%), oat (10–20.65%) barley (8–15%) grown in Canada [28,29,30,31]. Previous Canary seed studies have reported minor variations in Canary seed protein concentrations [8,32,33], suggesting stability in this trait. Despite the narrow range observed, analysis of variance (Table 2) determined that protein concentration was significantly (*p* < 0.05) affected by variation due to genotype but not growing year. In particular, CDC Lumio had the lowest mean among the four genotypes tested (*p* < 0.05). In contrast, the protein concentration in oat [34], wheat [35], rice [36], and barley [37] are affected by genotype, environment, and their interaction. Nutritionally, Canary seed protein has a comparable digestibility to wheat protein but a lower amino acid score of 24 (vs. 33 for wheat), with lysine being the limiting amino acid [32,33]. However, the digestibility of Canary seed protein has been shown to improve with thermal processing [38].

The oil content ranged from 8 to 11% (Table 1) and was not significantly affected by genotypic differences or growing years (Table 2). Canary seed oil consists of 57% polyunsaturated fatty acids (of which 96% is linoleic acid), 29% monosaturated fatty acid (oleic acid), and 12% saturated (of which 92% is palmitic acid) [32,39]. Thus among cereal grains, Canary seed oil content is within the range of oat (5–10%) [29,40] but higher than that of wheat, barley, maize, millet, rye, sorghum, and rice [41,42,43].

Starch content ranged from 56.3 to 61.2% and was significantly (*p* < 0.0001) affected by the growing years but not by genotypic differences or their interaction. Starch primarily consists of two glucose polymers (amylose and amylopectin). Amylose contains a linear (1–4)-α-glucan chain with minimal branches at α (1–6), whereas amylopectin is highly branched. The ratio of these two polymers affects both the nutritional and physical properties of starch [44]. As shown in Table 2, Canary seed amylose content ranged from 21.9 to 26.1% of the starch. Our results on amylose content are similar to those reported by Irani et al. [11]. Abdel-Aal et al. [45] found that amylose content varied from 16 to 19.5% depending on genotype and growing environment. Our results on amylose content for CDC Calvi and CDC Cibo were higher than those reported by Abdel-Aal et al. [10], which confirms that the growing environment is an essential factor in the amylose content of Canary seed. Analysis of variance (Table 2) suggests that the growing year explained ~68% of the variation in amylose content, with genotypic variation explaining ~27%. The ranges observed in our study for both starch and amylose are typical of non-waxy-cereal grains [9] but higher than those of bean [46] and lower than lentil and pea [47].

### 3.2. Dietary Fiber Composition

Total dietary fiber ranged from 7.2 to 8.5%, of which ~75% was water-insoluble (Table 3). Our results suggest that Canary seed grains have an equal proportion of high molecular weight soluble dietary fiber (~0.95%) and low molecular weight soluble dietary fiber (~0.92%) (Table 3). Beta-glucan constituted ~38.4% of the high molecular weight soluble dietary fiber. Variance analysis (Table 2) indicated that the total dietary fiber concentration was not affected by the genotype, growing year, or their interaction. Our results for the high molecular weight soluble and insoluble dietary fiber are similar to those reported in the literature [9,32]. However, our reported total dietary fiber is higher due to the inclusion of low molecular weight dietary fiber, which has not previously been reported in other studies. The results presented in Table 3 indicate that Canary seed contains about 2.3–2.8% non-starch polysaccharides (NSP). Arabinose, xylose, and uronic acid values suggest that Canary seed has about 1.3% arabinoxylans, representing ~51% of NSP. In addition, based on the glucose and beta-glucan values (Table 3), the Canary seed groat contains about 0.89% cellulose, accounting for 34% of the NSP. The differences in genotype (*p* < 0.001), growing year (*p* < 0.05), and their interaction (*p* < 0.05) significantly influenced the content of xylose in Canary seed (Table 2). Consequently, genotype and growing year significantly influenced total NSP concentration (*p* < 0.05).

### 3.3. Micronutrient Composition

The concentration of vitamin A measured as retinol equivalents was less than 20 IU/100 g, and that of vitamin E as alpha-tocopherol was less than 1 IU/100 g for all genotypes in two years (Table 4), suggesting that Canary seed is a poor source of fat-soluble vitamins. Other studies have reported that Canary seed contains 6.3–8.9 µg/g carotenoid content [48] and B vitamins such as thiamine (~0.85 mg/100 g), riboflavin (~0.16 mg/100 g), and niacin (0.89 mg/100 g) [8]. The mineral composition of Canary seed is presented in Table 4. Our results suggest that Canary seed is a good dietary source of magnesium, phosphorous, manganese, iron, zinc, and copper, based on a 40 g serving size for children and adults. Similar to other cereal grains, Canary seed is a poor source of sodium, calcium, and potassium. Other than iron, the concentration of the minerals in Canary seed was not influenced by the genotypic differences in this study (Table 2). Conversely, the growing year significantly affected the concentration of copper but not for the other minerals. Our results for manganese, magnesium, calcium, and potassium are similar to those reported in the literature [9], which supports independence against genotypic or environmental differences.

### 3.4. Mono-, Di-, and Oligosaccharide Composition

The results in Table 5 present the soluble sugar concentration levels in Canary seed. Among the screened mono- and disaccharides (fructose, glucose, maltose, and sucrose), only sucrose was found in detectable and quantifiable amounts (Figure 1). Sucrose, which ranged from 0.77 to 1.03% (Table 5), was significantly (*p* < 0.0001) affected by genotype, growing year, and their interaction (Table 2). Our results on sucrose are within the range of those reported by Abdel-Aal et al. [32]. The mean sucrose concentration of Canary seed from 2018 was 10% higher than that of 2019. The results also suggest that the sucrose concentration of Canary seed is similar to the sucrose concentration in other cereal grains, such as wheat, oat, and barley.

Among the two fermentable oligosaccharides screened, raffinose and stachyose, (Figure 1), only raffinose was present in detectable and quantifiable concentrations in all genotypes and years. Raffinose ranged from 0.3 to 0.5% (Table 5) and was significantly affected by genotype (*p* < 0.001), growing year (*p* < 0.05), and their interaction (*p* < 0.0001) (Table 2). Our results (Table 5) suggest that the raffinose content of Canary seed is comparable to that of wheat (0.326–0.70%), lower than that of barley (0.59–0.79%) and higher than in oat (< 0.08%) [49]. High levels of raffinose are not desirable despite its potential for probiotic properties, as it may cause flatulence in some consumers [50]. However, the concentration of raffinose family oligosaccharides present in cereal grains is too low to illicit noticeable flatulence.

In summary, the nutritional properties (relevant to nutritional labeling) of Canary seed flour (protein, fat, minerals, vitamins, and dietary fiber), other than the starch content, are very stable among the tested cultivars across different growing environments. Thus, our results suggest that the food industry’s quality control program for sourcing Canary seed as an ingredient should focus on starch content to ensure compliance with nutritional labeling guidelines. Also, there is a need for more research to understand the environmental factors affecting Canary seed flour’s starch content for its successful utilization as a novel ingredient.

**Table 4 foods-13-01786-t004:** Micronutrient composition of Canary seed grown in 2018 and 2019.

Year	Variety	Fat-Soluble Vitamin (IU/100 g)		Mineral Concentration (mg/100 g)								
		E (α-tocopherol)	A (Retinol)	Magnesium	Sodium	Potassium	Calcium	Phosphorus	Manganese	Iron	Copper	Zinc
2018	C10045	<1.0	<20.0	185.0 ± 1.1 ^a^	29.1 ± 6.8 ^a^	356.8 ± 0.1 ^a^	46.6 ± 1.1 ^a^	530.9 ± 15.2 ^a^	6.0 ± 0.2 ^a^	8.5 ± 0.2 ^bc^	1.5 ± 0.5 ^ab^	6.0 ± 0.6 ^a^
2018	CDC Lumio	<1.0	<20.0	181.1 ± 3.9 ^a^	20.2 ± 1.3 ^a^	351.3 ± 12.4 ^a^	42.9 ± 4.5 ^a^	507.8 ± 43.4 ^a^	6.0 ± 0.1 ^a^	8.0 ± 0.3 ^c^	0.9 ± 0.0 ^b^	6.3 ± 0.3 ^a^
2018	CDC Calvi	<1.0	<20.0	202.3 ± 2.1 ^a^	20.0 ± 3.5 ^a^	396.1 ± 28.9 ^a^	50.0 ± 3.8 ^a^	581.1 ± 53.7 ^a^	6.9 ± 0.3 ^a^	10.5 ± 0.3 ^a^	1.6 ± 0 ^ab^	7.6 ± 1.6 ^a^
2018	CDC Cibo	<1.0	<20.0	191.8 ± 2.5 ^a^	23.5 ± 4.8 ^a^	358.0 ± 1.4 ^a^	45.9 ± 3.2 ^a^	544.2 ± 47.0 ^a^	6.2 ± 0.3 ^a^	9.1 ± 0.5 ^abc^	1.3 ± 0 ^ab^	6.6 ± 0.2 ^a^
Average				190.1 ^A^	23.2 ^A^	365.5 ^A^	46.4 ^A^	541.0 ^A^	6.3 ^A^	9.0 ^A^	1.3 ^B^	6.6 ^A^
2019	C10045	<1.0	<20.0	181.2 ± 7.8 ^a^	16.4 ± 0.7 ^a^	331.3 ± 15.1 ^a^	41.1 ± 1.6 ^a^	493.5 ± 7.7 ^a^	5.3 ± 0.8 ^a^	8.8 ± 0.4 ^bc^	1.5 ± 0.5 ^ab^	6.2 ± 1.3 ^a^
2019	CDC Lumio	<1.0	<20.0	176.1 ± 4.7 ^a^	26.2 ± 5.1 ^a^	324.2 ± 0.1 ^a^	40.9 ± 4.0 ^a^	459.4 ± 9.9 ^a^	5.7 ± 0.7 ^a^	8.0 ± 0.5 ^c^	2.6 ± 0.1 ^ab^	6.1 ± 1.0 ^a^
2019	CDC Calvi	<1.0	<20.0	191.6 ± 3.1 ^a^	22.2 ± 0.8 ^a^	364.6 ± 16.0 ^a^	47.1 ± 3.7 ^a^	531.5 ± 8.0 ^a^	6.1 ± 0.7 ^a^	9.0 ± 0.2 ^abc^	1.5 ± 0.1 ^ab^	8.4 ± 1.7 ^a^
2019	CDC Cibo	<1.0	<20.0	196.8 ± 23.3 ^a^	18.8 ± 4.3 ^a^	370.4 ± 42.2 ^a^	41.8 ± 1.7 ^a^	565.9 ± 72.1 ^a^	5.9 ± 1.0 ^a^	10.0 ± 0.5 ^ab^	3 ± 1.2 ^a^	7.4 ± 2.1 ^a^
Average				186.5 ^A^	20.9 ^A^	347.6 ^A^	42.7 ^A^	512.6 ^A^	5.8 ^A^	9.0 ^A^	2.1 ^A^	7.0 ^A^
RDA		7.5–22.4	300–900	80–420	800–1500	2000–2600	700–1300	460–1250	1.2–2.3	7–18	0.34–0.9	3–11

Note: The data represent the mean ± standard deviation. Means were compared using the Tukey–Kramer HSD procedure of SAS 9.4 software. Means of each analyte not connected with the same lowercase superscript are significantly different (*p* ≤ 0.05). Year means of each analyte not connected with the same uppercase superscript in each column are significantly different (*p* ≤ 0.05). RDA means Recommended Daily Allowance and values are based on Canadian food guide document for children and adults.

**Table 5 foods-13-01786-t005:** Di- and oligosaccharide content of four genotypes of Canary seed grown in 2018 and 2019 growing years.

Genotype	Maltose, % db		Sucrose, % db		Raffinose, % db		Stachyose, % db	
	2018	2019	2018	2019	2018	2019	2018	2019
C10045	nd	nd	0.92 ± 0.01 ^ab^	0.77 ± 0.00 ^c^	0.42 ± 0.01 ^ab^	0.32 ± 0.00 ^b^	nd	nd
CDC Lumio	nd	nd	1.01 ± 0.00 ^a^	0.94 ± 0.02 ^ab^	0.37 ± 0.00 ^ab^	0.37 ± 0.05 ^ab^	nd	nd
CDC Calvi	nd	nd	0.96 ± 0.06 ^ab^	0.96 ± 0.02 ^ab^	0.41 ± 0.05 ^ab^	0.48 ± 0.01 ^a^	nd	nd
CDC Cibo	nd	nd	0.99 ± 0.01 ^a^	0.85 ± 0.01 ^bc^	0.45 ± 0.00 ^ab^	0.38 ± 0.02 ^ab^	nd	nd
Average	nd	nd	0.97 ± 0.03 ^A^	0.88 ± 0.08 ^B^	0.41 ± 0.03 ^A^	0.39 ± 0.06 ^A^	nd	nd

Note: The data represent the mean ± standard deviation. Means were compared using the Tukey–Kramer HSD procedure of SAS 9.4 software. Means of each analyte not connected with the same lowercase superscript are significantly different (*p* ≤ 0.05). Year means of each analyte not connected with the same uppercase superscript in each row are significantly different (*p* ≤ 0.05).

### 3.5. Effect of Genotype and Growing Year on Pasting Properties of Canary Seed

The pasting properties of Canary seed flour as measured in viscosity (peak, trough, breakdown, final, and setback), peak time, and pasting temperature are presented in Table 6. The RVA pasting curve (Figure 2) shows the changes in viscosity during the heating and cooling of the starch slurry under shear stress. Our results suggest that the mean minimum temperature required to cook Canary seed flour starch is 94.7 °C as determined by their pasting temperatures. The peak viscosity of oat, refined wheat, and whole wheat flour were 3.2-, 2.5-, and 1.6-times that of Canary seed flour, suggesting that Canary seed flour has a lower water holding capacity than wheat and oat flour. The analysis of variance (Table 2) suggests that the peak viscosity was significantly affected by genotype (*p* < 0.001) but not the growing year. The Pearson correlation analysis suggests that observed differences in peak viscosity of Canary seed flours may be related to their differences in amylose (R = 0.81, *p* < 0.0001) and starch (R = 0.69, *p* < 0.0031) content. Similarly, Yan et al. [51] also observed that the amylose content of sorghum starches influenced their pasting properties.

The hot paste viscosity under continued shear stress measured as trough viscosity varied greatly and was genotype- (*p* < 0.01) and growing year (*p* < 0.01)-dependent. Pearson correlation analysis showed that hot paste viscosity was significantly (R = −0.75, *p* < 0.0007) correlated to the L* color value of Canary seed flour. In particular, yellow Canary seed flour (CDC Cibo and C10045) had a significantly lower trough viscosity than brown Canary seed (CDC Lumio and CDC Calvi) flour. Consequently, the yellow Canary seed cultivars registered a ~30% breakdown in viscosity during heating and shearing compared to the brown cultivars at ~20%, indicating that brown cultivars had a higher stability to shear stress than yellow Canary seed flours.

During the cooling phase, starch polymers retrograde, which may contribute to product texture. The final viscosity of Canary seed flour, which indicates the ability of the flour to form a gel, was not significantly different (*p* < 0.05) across the tested varieties and growing years. Figure 2 shows that the final viscosity of Canary seed flour was comparable to refined wheat flour, higher than whole wheat flour, and lower than oat flour. Unlike the wheat and oat control samples tested, we observed that the Canary seed viscosity decreased during prolonged cooling and shearing. A desirable trait for industrial applications is stability in the pasting properties of an ingredient.

The presence of other compounds in Canary seed flour, such as proteins, lipids, and enzymes, affects the pasting properties of starch. Abdel-Aal et al. [39] reported pasting properties of Canary seed starches isolated from CDC Calvi (brown) and CDC Cibo (yellow). Like our results, they also observed that CDC Cibo, a yellow cultivar, had lower hot paste viscosity than CDC Calvi. Thus, for food applications with pasting being a key processing factor, genotype and growing environment should be considered when sourcing Canary seed as an ingredient to ensure consistency in processing and final product characteristics.

### 3.6. Effect of Genotype and Growing Year on Canary Seed Flour and Groat Color

The color measurements of Canary seed groat and flour are presented in Table 7. The difference in groat color was evident between yellow and brown genotypes, and insignificant differences were observed in L*, a*, and b* values within each color group. Yellow Canary seed genotypes had significantly higher L*, a*, and b* values than brown lines. However, after milling the groats, the color differences between brown and yellow genotypes were greatly reduced as the external color was diluted by starch. Still, flour obtained from yellow cultivars had significantly higher L*, a*, and b* values than brown cultivars. Analysis of variance (Table 2) indicated that the growing year was not a significant determinant of the color of the four Canary seed genotypes tested, as it only explained less than 5% of the observed color variance.

**Table 6 foods-13-01786-t006:** Pasting properties of four genotypes of Canary seed grown in 2018 and 2019 growing years.

		C10045	CDC Lumio	CDC Calvi	CDC Cibo	Mean
Peak, cP	2018	1072.0 ± 10.5 ^a^	1108.3 ± 6.8 ^a^	1104.0 ± 10.5 ^a^	1099.0 ± 14.5 ^a^	1095.8 ± 14.1 ^A^
	2019	1029.0 ± 29.5 ^a^	1047.3 ± 9.8 ^a^	1071.5 ± 22.5 ^a^	1068.0 ± 5.0 ^a^	1053.9 ± 17.1 ^B^
Trough, cP	2018	709.3 ± 9.8 ^b^	830.5 ± 9.0 ^ab^	830.5 ± 22.0 ^ab^	700.0 ± 15.0 ^b^	767.6 ± 63.0 ^B^
	2019	756.0 ± 32.0 ^ab^	878.5 ± 1.0 ^a^	893.5 ± 30.5 ^a^	829.5 ± 64.0 ^ab^	839.4 ± 53.6 ^A^
Breakdown, cP	2018	362.8 ± 0.8 ^ab^	277.8 ± 2.3 ^abc^	273.5 ± 11.5 ^abc^	399.0 ± 0.5 ^a^	328.3 ± 54.2 ^A^
	2019	273.0 ± 2.5 ^abc^	168.8 ± 8.8 ^c^	178.0 ± 8.0 ^c^	238.5 ± 69.0 ^b^	214.6 ± 43.1 ^B^
Final, cP	2018	2479.3 ± 87.8 ^a^	2596.0 ± 13.5 ^a^	2488.3 ± 116.8 ^a^	2401.5 ± 39.5 ^a^	2491.3 ± 69.3 ^A^
	2019	2692.3 ± 44.3 ^a^	2370.3 ± 30.8 ^a^	2392.3 ± 62.8 ^a^	2616.3 ± 157.3 ^a^	2517.8 ± 139.3 ^A^
Setback, cP	2018	1770.0 ± 78.0 ^ab^	1765.5 ± 22.5 ^ab^	1657.8 ± 94.8 ^ab^	1701.5 ± 54.5 ^ab^	1723.7 ± 46.7 ^A^
	2019	1936.3 ± 12.3 ^a^	1491.8 ± 29.8 ^b^	1498.8 ± 32.3 ^b^	1786.8 ± 93.3 ^ab^	1678.4 ± 190.6 ^A^
Peak time, min	2018	6.22 ± 0.02 ^cd^	6.45 ± 0.02 ^bcd^	6.48 ± 0.05 ^abc^	6.17 ± 0.03 ^d^	6.33 ± 0.14 ^B^
	2019	6.40 ± 0.04 ^bcd^	6.57 ± 0.00 ^ab^	6.75 ± 0.05 ^a^	6.48 ± 0.12 ^abc^	6.55 ± 0.13 ^A^
Pasting Temperature, °C	2018	94.7 ± 0.0 ^b^	94.6 ± 0.0 ^b^	94.7 ± 0.0 ^ab^	94.7 ± 0.0 ^b^	94.7 ± 0.0 ^B^
	2019	94.9 ± 0.1 ^a^	94.8 ± 0.1 ^ab^	94.9 ± 0.0 ^a^	94.9 ± 0.1 ^ab^	94.9 ± 0.0 ^A^

Note: The data represent the mean ± standard deviation. Means were compared using the Tukey–Kramer HSD procedure of SAS 9.4 software. Means of each pasting attribute not connected with the same lowercase superscript are significantly different (*p* ≤ 0.05). Year means of each pasting attribute not connected with the same uppercase superscript in each column are significantly different (*p* ≤ 0.05).

**Table 7 foods-13-01786-t007:** Color of groat and flour of four genotypes of Canary seed grown in 2018 and 2019 growing years.

Genotype		Color L*		Color a*		Color b*	
		2018	2019	2018	2019	2018	2019
Canary seed groats							
C10045	Yellow	52.10 ± 1.67 ^a^	47.8 ± 0.56 ^a^	7.00 ± 0.26 ^a^	7.35 ± 0.10 ^a^	21.63 ± 0.15 ^a^	22.09 ± 0.6 ^a^
CDC Lumio	Brown	39.55 ± 1.25 ^b^	36.98 ± 0.07 ^b^	3.41 ± 0.14 ^b^	3.18 ± 0.01 ^b^	9.06 ± 0.05 ^b^	8.44 ± 0.05 ^b^
CDC Calvi	Brown	41.21 ± 0.98 ^b^	38.48 ± 0.07 ^b^	3.53 ± 0.11 ^b^	3.56 ± 0.05 ^b^	9.95 ± 0.23 ^b^	10.38 ± 0.07 ^b^
CDC Cibo	Yellow	49.69 ± 0.25 ^a^	48.29 ± 1.13 ^a^	7.31 ± 0.01 ^a^	6.94 ± 0.07 ^a^	22.31 ± 0.19 ^a^	22.46 ± 0.99 ^a^
Average		45.64 ± 5.36 ^A^	42.89 ± 5.19 ^B^	5.31 ± 1.85 ^A^	5.26 ± 1.90 ^A^	15.74 ± 6.25 ^A^	15.84 ± 6.47 ^A^
Canary seed Flour							
C10045	Yellow	82.10 ± 0.57 ^a^	81.93 ± 0.20 ^a^	−0.80 ± 0.13 ^a^	−1.00 ± 0.08 ^a^	17.61 ± 0.06 ^bc^	18.66 ± 0.13 ^a^
CDC Lumio	Brown	77.48 ± 0.60 ^b^	76.61 ± 0.74 ^b^	−1.81 ± 0.11 ^b^	−1.71 ± 0.09 ^b^	12.87 ± 0.20 ^e^	13.64 ± 0.18 ^de^
CDC Calvi	Brown	78.19 ± 0.78 ^b^	77.60 ± 0.08 ^b^	−1.83 ± 0.07 ^b^	−1.73 ± 0.00 ^b^	13.08 ± 0.06 ^de^	13.80 ± 0.08 ^d^
CDC Cibo	Yellow	81.82 ± 0.29 ^a^	81.28 ± 0.15 ^a^	−0.69 ± 0.10 ^a^	−0.84 ± 0.02 ^a^	16.9 ± 0.06 ^c^	18.14 ± 0.33 ^ab^
Average		79.90 ± 2.08 ^A^	79.35 ± 2.29 ^A^	−1.28 ± 0.54 ^A^	−1.32 ± 0.41 ^A^	15.12 ± 2.16 ^B^	16.06 ± 2.35 ^A^

Note: The data represent the mean ± standard deviation. Means were compared using the Tukey–Kramer HSD procedure of SAS 9.4 software. Means of each analyte not connected with the same lowercase superscript are significantly different (*p* ≤ 0.05). Year means of each analyte not connected with the same uppercase superscript in each row are significantly different (*p* ≤ 0.05).

## 4. Conclusions

This research aimed to provide the effects of genotype and growing year on the nutritional and pasting properties of Canary seed flour as baseline information for the food industry. Canary seed flour is a novel food ingredient, and understanding the factors affecting its nutritional and technological properties is important for the food industry. Hairless (glabrous) Canary seed flour has high protein content but low fat-soluble vitamins. In general, the major nutritional components of Canary seed flour were not affected by genotype and growing year except for starch content, which was significantly affected by the growing year (*p* < 0.0001). Genotypic variation significantly affected the trough, breakdown, and setback viscosities and peak time of Canary seed flours. Conversely, the growing year significantly affected all of the pasting properties except final and setback viscosities. The limitation of our study is the sample size, as only four genotypes and one location were used. Overall, our results suggest that the food industry should strategize in sourcing Canary seed to ensure consistency in nutritional quality and performance in processing.

## Figures and Tables

**Figure 1 foods-13-01786-f001:**
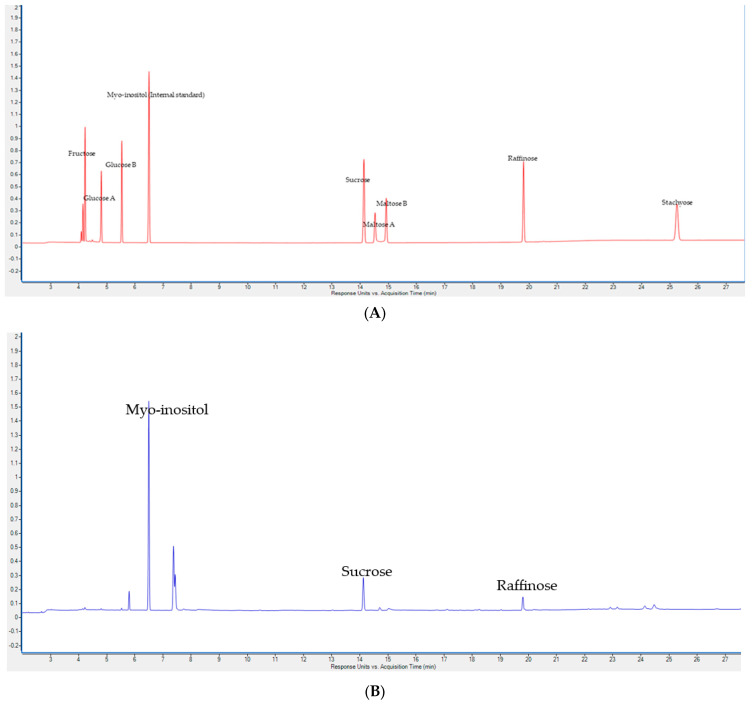
(**A**) Mono-, di-, and oligosaccharide standard chromatogram; (**B**) Canary seed mono-, di-, and oligosaccharide chromatogram.

**Figure 2 foods-13-01786-f002:**
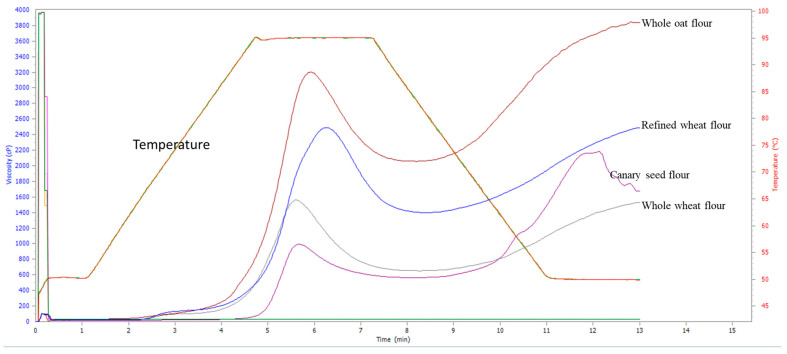
RVA pasting curve of Canary seed, wheat, and oat flours.

**Table 1 foods-13-01786-t001:** Macronutrient composition of four genotypes of Canary seed grown in 2018 and 2019 growing years.

Genotype	Color	Protein, % db		Oil, % db		Starch, % db		Amylose, % starch	
		2018	2019	2018	2019	2018	2019	2018	2019
C10045	Yellow	23.8 ± 0.8 ^a^	23.1 ± 0.3 ^a^	8.9 ± 0.8 ^a^	9.8 ± 0.3 ^a^	58.7 ± 0.3 ^abc^	56.9 ± 0.3 ^bc^	24.4 ± 0.3 ^c^	22.2 ± 0.2 ^e^
CDC Lumio	Brown	22.8 ± 0.1 ^a^	22.9 ± 0.8 ^a^	9.5 ± 0.1 ^a^	10.2 ± 0.8 ^a^	59.0 ± 0.7 ^abc^	56.8 ± 0.5 ^c^	26.1 ± 0.0 ^a^	23.6 ± 0.1 ^cd^
CDC Calvi	Brown	23.9 ± 0.2 ^a^	23.5 ± 0.1 ^a^	9.2 ± 0.2 ^a^	9.1 ± 0.1 ^a^	60.1 ± 1.0 ^a^	56.7 ± 0.1 ^c^	25.6 ± 0.1 ^ab^	23.7 ± 0.2 ^cd^
CDC Cibo	Yellow	23.2 ± 0.3 ^a^	23.6 ± 0.1 ^a^	10.0 ± 0.3 ^a^	9.2 ± 0.1 ^a^	59.7 ± 0.2 ^ab^	56.8 ± 0.3 ^bc^	24.6 ± 0.1 ^bc^	23.3 ± 0.1 ^d^
Average		23.4 ± 0.4 ^A^	23.3 ± 0.3 ^A^	9.4 ± 0.4 ^A^	9.6 ± 0.4 ^A^	59.4 ± 0.5 ^A^	56.8 ± 0.1 ^B^	25.2 ± 0.7 ^A^	23.2 ± 0.6 ^B^

Note: The data represent the mean ± standard deviation. Means were compared using the Tukey–Kramer HSD procedure of SAS 9.4 software. Means of each analyte not connected with the same lowercase superscript are significantly different (*p* ≤ 0.05). Year means of each analyte not connected with the same uppercase superscript in each row are significantly different (*p* ≤ 0.05).

**Table 2 foods-13-01786-t002:** Analysis of variance for the effect of cultivar and growing year on nutritional composition, color, and pasting properties of Canary seed.

	Genotype (G)	Year (Y)	G X Y		Genotype (G)	Year (Y)	G X Y
Degrees of Freedom	3	1	3		3	1	3
Protein	5.79 * (48.6)	1.31 (3.7)	3.02 (25.4)	Beta-glucan	2.92 (10.4)	67.24 **** (79.6)	0.15 (0.5)
Oil	0.96 (17.7)	0.39 (2.4)	1.67 (30.8)	TDF	2.77 (23.3)	2.48 (6.9)	0.32 (2.7)
Starch	0.62 (2.9)	50.73 **** (80.2)	0.88 (4.2)	IDF	1.55 (14.6)	0.37 (1.2)	0.90 (8.6)
Amylose	31.72 **** (26.7)	243.37 **** (68.2)	3.48 (2.9)	SDF	1.24 (11.8)	1.79 (5.7)	0.67 (5.9)
Sucrose	29.18 **** (42.3)	67.66 **** (32.7)	9.26 *** (13.4)	LMWDF	0.54 (5.6)	0.03 (0.1)	1.17 (12.0)
Raffinose	9.55 *** (31.7)	5.15 * (5.7)	10.84 **** (36.0)	Arabinose	4.29 * (39.8)	2.56 (7.9)	2.96 (27.5)
Groat color L*	84.54 **** (90.1)	17.47 *** (6.2)	0.82 (0.9)	Xylose	10 ** (49.2)	8.13 * (13.3)	4.93 * (24.3)
Groat color a*	652.98 **** (99.0)	0.39 (0.0)	3.63 (0.6)	Glucose	2.49 (23.8)	7.18 * (22.9)	2.88 (27.6)
Groat color b*	590.82 **** (99.4)	0.12 (0.0)	0.68 (0.1)	Glucuronic acid	1.68 (37.4)	0.02 (0.2)	0.14 (3.1)
Flour color L*	50.98 **** (93.4)	2.37 (1.4)	0.16 (0.3)	NSP	4.97 * (32.7)	10.48 * (23)	4.07 (26.8)
Flour color a*	81.16 **** (94.8)	0.36 (0.1)	1.64 (1.9)	Magnesium	3.07 (48.5)	0.52 (2.7)	0.42 (6.7)
Flour color b*	502.45 **** (95.1)	66.17 **** (4.2)	1.12 (0.2)	Sodium	0.29 (3.8)	1.32 (5.8)	4.22 * (55.4)
Peak Viscosity	2.28 (22.6)	14.22 ** (47.1)	0.39 (3.8)	Potassium	3.72 (44)	3.16 (12.4)	1.02 (12)
Trough viscosity	9.65 ** (56.2)	11.90 ** (23.1)	0.88 (5.2)	Calcium	3.15 (40.5)	5.17 (22.2)	0.24 (3)
Breakdown Viscosity	9.30 ** (35.2)	40.98 *** (51.6)	0.82 (3.1)	Phosphorus	3.17 (43.7)	2.05 (9.4)	0.73 (10)
Final Viscosity	1.09 (14.7)	0.21 (0.9)	3.63 (48.7)	Manganese	1.84 (32.4)	2.94 (17.2)	0.2 (3.5)
Setback	8.24 ** (48.3)	1.12 (2.2)	5.77 * (33.8)	Iron	17.63 *** (65.1)	0.18 (0.2)	6.7 * (24.8)
Peak time	16.29 *** (50.3)	36.02 *** (37.1)	1.44 (4.5)	Copper	1.38 (11.9)	10.93 * (31.4)	3.92 (33.8)
Pasting Temperature	2.06 (9.2)	51.82 **** (76.9)	0.45 (2.0)	Zinc	1.88 (39)	0.42 (2.9)	0.13 (2.8)

Note: Data show F values and asterisk means significant level as follows: *, **, ***, and **** means *p* value = <0.05, <0.01, <0.001, and <0.0001, respectively. The data in brackets are percent contribution to the variation.

**Table 3 foods-13-01786-t003:** (**A**) Dietary fiber composition of four genotypes of Canary seed grown in 2018 and 2019 growing years. (**B**) Non-starch polysaccharide composition of four genotypes of Canary seed grown in 2018 and 2019 growing years.

(A)									
Genotype		IDF, % db		HMWSDF, % db		LMWSDF, % db		TDF, % db	
		2018	2019	2018	2019	2018	2019	2018	2019
C10045		6.0 ± 0.2 ^a^	5.8 ± 0.0 ^a^	0.8 ± 0.0 ^a^	1.0 ± 0.0 ^a^	1.0 ± 0.1 ^a^	0.8 ± 0.1 ^a^	7.8 ± 0.0 ^a^	7.6 ± 0.1 ^a^
CDC Lumio		5.8 ± 0.1 ^a^	5.9 ± 0.1 ^a^	0.8 ± 0.0 ^a^	1.1 ± 0.2 ^a^	1.0 ± 0.1 ^a^	0.9 ± 0.0 ^a^	7.5 ± 0.0 ^a^	7.8 ± 0.1 ^a^
CDC Calvi		6.0 ± 0.2 ^a^	6.1 ± 0.1 ^a^	1.2 ± 0.2 ^a^	1.1 ± 0.3 ^a^	0.9 ± 0.1 ^a^	0.9 ± 0.2 ^a^	8.1 ± 0.0 ^a^	8.1 ± 0.1 ^a^
CDC Cibo		5.8 ± 0.1 ^a^	5.9 ± 0.2 ^a^	0.7 ± 0.1 ^a^	1.0 ± 0.2 ^a^	1.1 ± 0.3 ^a^	0.9 ± 0.1 ^a^	7.6 ± 0.3 ^a^	7.8 ± 0.3 ^a^
Average		5.9 ± 0.1 ^A^	5.9 ± 0.1 ^A^	0.9 ± 0.2 ^A^	1.1 ± 0.0 ^A^	1.0 ± 0.1 ^A^	0.9 ± 0.0 ^A^	7.7 ± 0.2 ^A^	7.9 ± 0.2 ^A^
**(B)**									
		**Arabinose, % db**	**Xylose, % db**	**Glucose, % db**	**Uronic Acid, % db**	**NSP, % db**		**Beta Glucan, % db**	
2018	C10045	0.48 ± 0.00 ^a^	0.58 ± 0.01 ^abc^	1.32 ± 0.01 ^a^	0.30 ± 0.02 ^a^	2.68 ± 0.15 ^ab^		0.39 ± 0.01 ^abc^	
2018	CDC Lumio	0.42 ± 0.04 ^a^	0.52 ± 0.05 ^bc^	1.29 ± 0.01 ^a^	0.32 ± 0.01 ^a^	2.55 ± 0.09 ^ab^		0.41 ± 0.01 ^a^	
2018	CDC Calvi	0.50 ± 0.00 ^a^	0.65 ± 0.03 ^a^	1.33 ± 0.06 ^a^	0.29 ± 0.03 ^a^	2.78 ± 0.00 ^a^		0.39 ± 0.01 ^ab^	
2018	CDC Cibo	0.46 ± 0.04 ^a^	0.56 ± 0.02 ^abc^	1.34 ± 0.22 ^a^	0.31 ± 0.01 ^a^	2.67 ± 0.28 ^ab^		0.38 ± 0.01 ^abcd^	
	Average	0.47 ± 0.02 ^A^	0.58 ± 0.02 ^A^	1.32 ± 0.10 ^A^	0.31 ± 0.02 ^A^	2.67 ± 0.13 ^A^		0.39 ± 0.01 ^A^	
2019	C10045	0.38 ± 0.00 ^a^	0.48 ± 0.00 ^c^	1.04 ± 0.02 ^a^	0.31 ± 0.00 ^a^	2.20 ± 0.02 ^b^		0.33 ± 0.01 ^abc^	
2019	CDC Lumio	0.45 ± 0.02 ^a^	0.56 ± 0.00 ^abc^	1.28 ± 0.10 ^a^	0.31 ± 0.01 ^a^	2.61 ± 0.11 ^ab^		0.35 ± 0.01 ^bcd^	
2019	CDC Calvi	0.50 ± 0.05 ^a^	0.60 ± 0.02 ^ab^	1.35 ± 0.03 ^a^	0.29 ± 0.02 ^a^	2.74 ± 0.05 ^a^		0.34 ± 0.01 ^cd^	
2019	CDC Cibo	0.43 ± 0.05 ^a^	0.51 ± 0.05 ^bc^	1.07 ± 0.03 ^a^	0.30 ± 0.01 ^a^	2.31 ± 0.07 ^ab^		0.33 ± 0.01 ^d^	
	Average	0.44 ± 0.03 ^A^	0.54 ± 0.02 ^B^	1.19 ± 0.05 ^B^	0.30 ± 0.01 ^A^	2.47 ± 0.07 ^B^		0.34 ± 0.01 ^B^	

Note: The data represent the mean ± standard deviation. Means were compared using the Tukey–Kramer HSD procedure of SAS 9.4 software. Means of each analyte not connected with the same lowercase superscript are significantly different (*p* ≤ 0.05). Year means of each analyte not connected with the same uppercase superscript in each row/column are significantly different (*p* ≤ 0.05).

## Data Availability

The original contributions presented in the study are included in the article, further inquiries can be directed to the corresponding authors.

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
