# Peer review of "Effects of Genotype and Growing Year on the Nutritional Composition and Pasting Properties of Glabrous Canary Seed (Phalaris canariensis L.) Groat Flours"

_foods, 2024, doi:10.3390/foods13111786_

Round 1

Reviewer 1 Report

Comments and Suggestions for Authors

The manuscript contains new information about an underexploited cereal in food processing. However, there are some fundamental mistakes, particularly in the carbohydrate classification.

In Table 3B, the non-starch polysaccharide content of four genotypes of Canary seed grown in 2018 and 2019 includes arabinose, xylose, glucose, uronic acid, NSP, and beta-glucan.

In Table 5, the simple sugar composition of four genotypes of Canary seed grown in 2018 and 2019

- includes: maltose, sucrose, raffinose, and stachyose.

Figure 1a shows the standard chromatogram for simple sugars

- includes: glucose, fructose, sucrose, raffinose, stachyose, maltose B, and maltose A.

Figure 1b displays the simple sugars chromatogram for Canary seed

- includes: sucrose and raffinose.

It is well known from the literature that terminology in carbohydrate classification is not universal, and the presence of variants complicates this issue. I suggest correcting and making the carbohydrate classification uniform throughout the manuscript. It is best to avoid the term "simple sugars" as it can confuse. Instead, it is preferable to use the terms mono-, di-, oligo-, and polymeric carbohydrates.

Due to their prebiotic function, oligo-saccharides are often considered as part of dietary fiber, not simple sugars.

Author Response

Response:

Thank you for your comment. Table 3B introduced the content and composition of the non-starch polysaccharide of Canary seed. The monosaccharide listed in the table 3b were obtained after acid hydrolysis of the NSP. Whereas, Table 5 shows free sugars (mono-, di-, oligo-saccharides) present in the flours. Fig 1a was meant to demonstrate that we screened for glucose, fructose, sucrose, raffinose, stachyose, and maltose. Out of which, only sucrose and raffinose were present in detectable and quantifiable concentration. Therefore, to improve clarity we have changed the titles accordingly to reflect the suggested terminology as follows:

Table 3B: Non-starch polysaccharide composition of four genotypes of Canary seed grown in 2018 and 2019 growing years

Table 5: Di- and oligo-saccharide content of four genotypes of Canary seed grown in 2018 and 2019 growing years

Figure 1: a) Mono-, di-, and oligosaccharide standard chromatogram

Figure 1: b) Canary seed’s mono-, di-, and oligosaccharide chromatogram

Reviewer 2 Report

Comments and Suggestions for Authors

The effect of genotype and growing environment on the nutritional composition, pasting properties, and color of Glabrous Canary seed groat flours were studied. They also compared the nutrient content of Glabrous Canary seed groat flour with other common cereal grains. The authors have done a lot of work. These comprehensive results would give fundamental information for food manufacturers, especially when developing some foods with special nutritional requirements. The merits and novelty of this study meet the requirements of Foods, and the study was well designed. Some questions in the manuscript need to be improved.

  1. The title can not reflect the content of this study accurately. 
  2. The genotype and growing environment may influence the nutritional composition and pasting properties of Glabrous Canary seed groat flours is the base of this study. The authors should add literature relative to the information reported before. If no result has been reported with Glabrous Canary seed groat flour, you can literature on the influence of genotype and growing environment on other cereals.
  3. There are some typing errors. For example, part of the capitalized letter in L40 should be shown in lowercase. And one of the "reported" should be removed.
  4. It is better to separate 2.2 into several parts according to macronutrient, micronutrient, dietary fiber, etc., to make the structure of this article more clear.
  5. Analysis of variance for the effect of cultivar and growing year in Table 2 should discussed at the end of this study because it needs to use data determined and discussed below. It doesn't follow conventional logic in its present form. Or, the authors should separate data in Table 2 to other tables to make the results more clear.
  6. I doubt the applicability of the method used for the determination of soluble sugar content. Maybe other disaccharides other than sucrose would also be detectable if other methods or dilutions were used.   

Author Response

1. The title cannot reflect the content of this study accurately. 

Response: Thank you for your comment. We have replaced environment with growing year to be more specific in the title

2. The genotype and growing environment may influence the nutritional composition and pasting properties of Glabrous Canary seed groat flours is the base of this study. The authors should add literature relative to the information reported before. If no result has been reported with Glabrous Canary seed groat flour, you can literature on the influence of genotype and growing environment on other cereals.

Response: Thank you comment. Yes limited data is found on GxE on Canary seed. We have now incorporated some reference with regards to other cereal grains.

3. There are some typing errors. For example, part of the capitalized letter in L40 should be shown in lowercase. And one of the "reported" should be removed.

Response: Thank you for your observation. We have put GRAS in brackets (L40) and deleted reported on L50.

4. It is better to separate 2.2 into several parts according to macronutrient, micronutrient, dietary fiber, etc., to make the structure of this article more clear.

Response: Thank you for your suggestions. We have separated the method and result section into subsections as suggested

5. Analysis of variance for the effect of cultivar and growing year in Table 2 should discussed at the end of this study because it needs to use data determined and discussed below. It doesn't follow conventional logic in its present form. Or, the authors should separate data in Table 2 to other tables to make the results more clear.

Response: Thank you for your suggestions. We followed the convention that requires tables to be listed as they appear in the manuscript. Splitting table 2 would result in un-manageable number of tables. In agreement with you, throughout discussion section, data is presented first before ANOVA is discussed.

6. I doubt the applicability of the method used for the determination of soluble sugar content. Maybe other disaccharides other than sucrose would also be detectable if other methods or dilutions were used.  Response: The method used to determine soluble mono, di- and oligosaccharides in cereal grains and fruits  by several workers in literature. Yes. It is very much possible that other sugars may be present but cannot be detected by the method used. Hence, our disclaimer in L 283 -284 stating that “Among the screened mono and disaccharides (fructose, glucose, maltose, and sucrose), only sucrose was found in detectable and quantifiable amounts (Figure 1)”. This statement recognizes that it is probable that other sugars may be present.